# Virus Quasispecies Rarefaction: Subsampling with or without Replacement?

**DOI:** 10.3390/v16050710

**Published:** 2024-04-29

**Authors:** Josep Gregori, Marta Ibañez-Lligoña, Sergi Colomer-Castell, Carolina Campos, Josep Quer

**Affiliations:** 1Liver Diseases-Viral Hepatitis, Liver Unit, Vall d’Hebron Institut de Recerca (VHIR), Vall d’Hebron Hospital Universitari, Vall d’Hebron Barcelona Hospital Campus, Passeig Vall d’Hebron 119-129, 08035 Barcelona, Spain; marta.ibanez@vhir.org (M.I.-L.); sergi.colomer@vhir.org (S.C.-C.); carolina.campos@vhir.org (C.C.); 2Centro de Investigación Biomédica en Red de Enfermedades Hepáticas y Digestivas (CIBERehd), Instituto de Salud Carlos III, Av. Monforte de Lemos, 3-5, 28029 Madrid, Spain; 3Biochemistry and Molecular Biology Department, Universitat Autònoma de Barcelona (UAB), Campus de la UAB, Plaça Cívica, 08193 Bellaterra, Spain; 4Medicine Department, Universitat Autònoma de Barcelona (UAB), Campus de la UAB, Plaça Cívica, 08193 Bellaterra, Spain

**Keywords:** rarefaction, haplotypes, quasispecies, metagenomics, subsampling

## Abstract

In quasispecies diversity studies, the comparison of two samples of varying sizes is a common necessity. However, the sensitivity of certain diversity indices to sample size variations poses a challenge. To address this issue, rarefaction emerges as a crucial tool, serving to normalize and create fairly comparable samples. This study emphasizes the imperative nature of sample size normalization in quasispecies diversity studies using next-generation sequencing (NGS) data. We present a thorough examination of resampling schemes using various simple hypothetical cases of quasispecies showing different quasispecies structures in the sense of haplotype genomic composition, offering a comprehensive understanding of their implications in general cases. Despite the big numbers implied in this sort of study, often involving coverages exceeding 100,000 reads per sample and amplicon, the rarefaction process for normalization should be performed with repeated resampling without replacement, especially when rare haplotypes constitute a significant fraction of interest. However, it is noteworthy that different diversity indices exhibit distinct sensitivities to sample size. Consequently, some diversity indicators may be compared directly without normalization, or instead may be resampled safely with replacement.

## 1. Introduction

The study of viral quasispecies diversity is similar in many ways to the study of biodiversity in ecology. Nevertheless, the size of a quasispecies is far beyond the size of any known ecosystem; for example, the number of viral particles in a patient chronically infected by the hepatitis C virus (HCV) may outnumber human population. Indeed, the dynamics in a quasispecies have no comparison with the dynamics in any ecosystem. Because of the low polymerase fidelity characteristic in these viruses, each viral replication cycle generates new variants [1,2,3]. With high viral loads, the number of viral particles generated and eliminated daily may be over 10^12^ [4,5].

The study of quasispecies diversity and composition through NGS by amplicons constitutes a powerful approach to this extremely high-diversity world, which nevertheless still falls short despite the developments since the times when molecular cloning was the only available tool to dig into quasispecies [6,7]. The study by amplicon haplotypes starts with the processing of next-generation sequencing (NGS) data to obtain a set of haplotypes and corresponding frequencies as read counts [8]. The depth of the analysis depends on the library size, that is, the number of sequenced reads by sample and amplicon. Groups of samples typically have different library sizes for technical variations, and library size normalization is required for fair comparisons, given that some diversity indices are highly sensitive to sample size. Rarefaction is a widely used normalization technique that involves random subsampling of reads from the initial sample library to a common library size. In the field of metagenomics, there is an open debate [9,10,11,12,13] about whether this process should be used at all. 

The basis of such debate is that by subsampling, we are increasing the downward bias already existing in our data. Bias exists in the sense that, first, we may only approach the true diversity of a microbiome, and second, that the more we limit our library size, the less representative it is of the studied population. Nevertheless, despite these limitations, it continues to be widely used in practice as a suitable normalization whenever diversity comparisons are required. It helps reduce the impact of an uneven sampling effort by eliminating the differential bias associated with more deeply sequenced samples. In metagenomic studies, the estimation of richness, which is the number of species identified in a sample, is paramount, and alternative approximations to estimate the true value, including unobserved species [14], might be preferred. In quasispecies studies, richness is understood as the number of observed and unobserved haplotypes, which plays a minor role, with other diversity computations being potentially preferable [15].

This report aims at clarifying the use of this essential tool in quasispecies diversity comparisons, the process of rarefaction, by which two samples of different sizes become comparable. In this context, rarefaction only makes sense in the frame of sample comparisons or when studying richness rarefaction curves of single samples. In ecology, rarefaction is defined as repeated resampling without replacement to the reference size. In the context of rarefaction, resampling without replacement implies that during a cycle of random subsampling, the extracted reads are not reintroduced into the initial sample pool. Conversely, resampling with replacement involves returning the sampled reads to the population after each random extraction. From a computational perspective, when dealing with substantial sample sizes, sampling with replacement tends to be faster than sampling without replacement. In a recent work with metagenomic data [11], the authors observed that rarefying libraries with or without replacement had no substantial impact on Shannon entropy. However, libraries rarefied with replacement exhibited a slightly reduced Shannon index compared to those rarefied without replacement across different library sizes. This effect is attributed to the exclusion of rare sequences, which occurred more frequently in sampling with replacement than in sampling without replacement. This suggests that samples dominated by only a few highly abundant sequence variants are comparatively robust to subsampling. Nevertheless, the authors stated that rarefying without replacement should be encouraged because it is more theoretically correct, specifically when representing the data so they account for the limitations of smaller library sizes [11]. We aim to study the impact of both sampling schemes with the large numbers implied in quasispecies analysis, both theoretically and numerically. 

Subsampling with replacement is equivalent to sampling from a population where haplotype frequencies remain constant. Using the cumulative distribution derived from these frequencies, extracting an item involves generating a random value from a uniform distribution. This value is then matched against the cumulative distribution to determine the corresponding haplotype ID. In this context, the item sampled with replacement represents a haplotype ID. In contrast, when sampling without replacement, the sampled item is a single read from the original sample. To prevent repetitions in successive random extractions, it becomes necessary to track which reads have been sampled and which have not.

## 2. Methods

Intensive resampling simulations under each hypothetical case have been carried out to support and extend the theoretical considerations for each case and resampling scheme. See Box 1 for definitions. Each described example is represented as a vector of reads, where each element corresponds to a different haplotype, and the total coverage is given by the sum of the vector elements. Based on this representation of quasispecies composition for each case, all the computations were conducted using R [16], with the help of packages knitr [17], tidyverse [18], ggplot2 [19], and dqrng [20]. Resampling without replacement was optimized using the dqsample.int function from the package dqrng [20], and with the help of a full sample size, vector mapping reads to haplotypes. The full R code used is given in Appendix A.

Box 1Rarefaction related definitions [15,21,22].

**Concept**

**Definition**
RarefactionA technique used to compensate for different intensities of sampling in diversity studies.Subsampling cycleIt consists in the successive random extraction of a given number of items from a sample, lower than the sample size, with or without replacement at each extraction.Subsampling with replacementThis
is based on a situation where an element is randomly extracted from a sample, identified, and then immediately replaced. Therefore, this element can be obtained again in further extractions along the same subsampling cycle.Subsampling without replacementAll extractions in a subsampling cycle are performed without replacement, so no item may be extracted multiple times in the same cycle.Downward biasInaccuracy in measurement or estimation that underestimates the true value.Subsampling fractionFraction of reads being subsampled from a given sample in a single resampling cycle.GranularityLevel of resolution at which the data are processed when estimating frequencies from counts.


## 3. Results

In the following sections, we study different specific cases to compose a comprehensive picture of the type of results to be expected with the two sampling schemes according to quasispecies structure. We start by showing the limitations inherent to sampling and subsampling with replacement. The following cases are studied under both subsampling schemes:All singletons: This represents a quasispecies where all haplotypes are represented by a single read. It serves as the simplest case to numerically show the discussed limitations, showcasing the most significant differences between the sampling schemes.Single dominant case: This hypothetical scenario involves a dominant haplotype, while all other haplotypes are singletons. Our goal is to evaluate the master frequency and the number of haplotypes.Prominent haplotypes: In this case, there are six prominent haplotypes along with a set of singletons. The objective is to evaluate the frequencies of the prominent haplotypes, the fraction of singletons in the quasispecies, and the fraction of reads for haplotypes with over one read and below the top 6 haplotypes, representing singleton replicates produced in sampling with replacement.No rare haplotypes: This is a quasispecies composed of a master haplotype at 90%, with 10 other haplotypes at 1% each. This scenario excludes singletons and lower frequency haplotypes. We seek to estimate haplotype frequencies by repeated subsampling.Flat quasispecies: Similar to the first case, all the haplotypes have equal frequencies, ranging from 1 read to 10 reads each, representing a perfectly even quasispecies. This case is crucial for demonstrating the robustness in sampling quasispecies data that have undergone a previous abundance filter at a low level.

Finally, we discuss the sensitivity of various diversity indices with respect to sample size, and the corresponding granularity caused by estimating frequencies from the counts.

### 3.1. Bootstrap: The Theory around 0.632

Given a sample of n items (reads), the probability to randomly extract any given item is 1/n, and the probability to not extract it is 1 − 1/*n*. In a bootstrap cycle composed of n extractions, each extraction is followed by a replacement, which makes successive extractions independent and with identical probability.

As a bootstrap cycle is composed of n extractions, the probability to not extract a given item from the sample in a full cycle is (1 − 1/*n*)*^n^*; this means that the probability to have a given item sampled in a bootstrap cycle is 1 − (1 − 1/*n*)*^n^*. As n tends to infinity, the limit of this expression is 1 − 1/*e* = 0.632. This result implies that a bootstrap resample is composed of 0.6321 unique realizations of items in the original sample plus 0.3679, which are replicates, in the limit as n grows to infinity.

### 3.2. Subsampling a Given Fraction with Replacement 

In subsampling with replacement to a given fraction of the sample size *f*, the number of random extractions in each cycle is *f · n*; then, the probability to have a given item in the subsample is 1 − (1 − 1*/n*)*^f·n^*, and the limit as n tends to infinity is 1 − (1*/e*)*^f^*.

To solve the limit, we transform the indetermination 1^∞^ to ∞ · 0, noting that *f (x) = e^ln(f(x))^*
$$\underset{n\to \infty }{lim}{\left(1-1/n\right)}^{f\cdot n}=\underset{n\to \infty }{lim}e^{f\cdot n\cdot ln(1-1/n)}$$

Next, ∞ · 0 is converted to 0/0 by double inversion of one term
$$\underset{n\to \infty }{lim}f\cdot n\cdot ln(1-1/n)=\underset{n\to \infty }{lim}f\cdot \frac{ln(1-1/n)}{1/n}$$
and finally, by the application of L’Hopital rule, differentiating the numerator and denominator:$$\underset{n\to \infty }{lim}f\cdot \frac{ln(1-1/n)}{1/n}=\underset{n\to \infty }{lim}f\cdot \frac{1/(1-1/n)\cdot 1/n^{2}}{-1/n^{2}}=\underset{n\to \infty }{lim}f\cdot \frac{-1}{\left(1-1/n\right)}=-f$$

So that
$$\underset{n\to \infty }{lim}{\left(1-1/n\right)}^{f\cdot n}=\underset{n\to \infty }{lim}e^{f\cdot n\cdot ln(1-1/n)}=e^{-f}={\left(\frac{1}{e}\right)}^{f}$$

Given the limit, Table 1 and Figure 1 show the expected fraction of items seen from the original sample in a resampling cycle at various fractions of the original size, from 0.1 to 1. In resampling without replacement, the fraction of seen items would correspond exactly to the fraction of subsampling. As the sample fraction increases, the deviation of seen items with respect to the sampled fraction also increases to reach its maximum at *f* = 1, corresponding to the pure bootstrap. This holds particular importance when studying richness by subsampling. Due to the replacement, some rare species may be observed with inflated frequencies in a subsampling cycle, while others may not be observed at all. This inflation adversely affects the representation of other rare species in the sampled data, as these will be sampled less.

### 3.3. The All-Singletons Case

The numerical approach involves creating a quasispecies sample composed of 10,000 unique reads, each representing a distinct haplotype with a single sequence. All of them are rare haplotypes. In this scenario, sampling without replacement will produce a resample with as many haplotypes as the resample size. Resampling with replacement will suffer from the 0.632 effect described above. In repeated subsampling without replacement to a fraction *f*, the number of haplotypes obtained is equal to the subsample size, as denoted in the ‘True’ column in Table 2.

Table 2 shows the results of *B* = 500 repeated resampling cycles at increasing fractions from 0.1 to 1, where ‘True’ is the true number of haplotypes in a fraction of the sample. ‘Expected’ is the number of expected haplotypes from the computed probability, ‘Median’ is the median number of haplotypes observed after *B* cycles of resampling with replacement, ‘Unique’ is the proportion of haplotypes observed as singletons, and ‘Replicated’ is the proportion of reads corresponding to replicated singletons.

A first conclusion is that in this case, an accurate estimate of richness is obtained exclusively when subsampling without replacement. On the other hand, the deviation from the true value diminishes proportionally with the decreasing fraction of subsampling.

### 3.4. The Single-Dominant Case

Let us consider a quasispecies composed of a dominant haplotype at varying frequencies, from 10% to 90%, with the remaining reads attributed to singletons. Our focus is on understanding how the estimation of the number of haplotypes and the frequency of the master haplotype unfolds through repeated subsampling, both with and without replacement. In this particular example, each quasispecies sample comprises 100,000 reads, with a master haplotype spanning from 10 to 90%. The remaining haplotypes in the sample are all singletons. Table 3 shows the quasispecies IDs and compositions.

Table 4 and Figure 2 show the results in estimating the number of haplotypes at different subsampling fractions after B resampling cycles with and without replacement, and Table 5 and Figure 3 show the results in estimating master frequencies.

As observed in the case of all singletons, the number of haplotypes is underestimated with respect to the true value when subsampling with replacement, contrary to the estimation by subsampling without replacement, which is very close to the true value. 

The estimation of the master haplotype frequency was nearly identical under both sampling schemes (Table 5 and Figure 3), contrary to the estimation of the number of haplotypes, which was severely downward biased when sampling with replacement, except for the lowest fractions of subsampling. 

This observation aligns with the data presented in Table 2 in the all-singletons case. An accurate estimation of the master frequency in this scenario implies that the complementary, which is the fraction of reads in the quasispecies for non-dominant haplotypes, is also accurate. In simpler terms, despite the underestimation in the number of haplotypes, the aggregated sum for non-dominant reads remains accurate. 

These results can be extended to say that prominent haplotypes will be accurately subsampled under both schemes, but rare haplotypes will be severely underestimated when subsampling with replacement, particularly when the fraction of reads for rare haplotypes in the sample is significant. 

Nonetheless, the estimate of the fraction of reads for low-abundance haplotypes may still be accurate.

### 3.5. Prominent Haplotypes

Let us consider now a quasispecies featuring six prominent haplotypes, each at half frequency of the preceding one, with the remaining haplotypes being singletons (Table 6). In this case, our goal is to estimate the frequencies of the six prominent haplotypes and to also determine the fraction of reads belonging to singleton haplotypes.

To account for singleton replicates, we compute the fraction of reads for haplotypes above 1 read but below the top 6 haplotypes in each subsampling.

Further analysis confirms that subsampling without replacement provides results very close to the true values. In contrast, subsampling with replacement estimates the frequencies of the prominent haplotypes fairly well, but underestimates the fraction of singletons, which is underestimated in favor of replicated singletons, with frequencies above 1 read and below the top 6 haplotypes. However, the sum of the fractions for singletons and replicated singletons, as the complementary to the top 6, remains well approximated, like in the single-dominant case (Table 7 and Table 8).

In conclusion, the frequencies of prominent haplotypes will be equally estimated under both schemes, but the number of rare haplotypes could be severely underestimated when subsampling with replacement. The aggregation of non-dominants is well approximated when subsampling with replacement. We observed similar results with real hepatitis E virus data [23] sequencing sample replicates at different depths [24].

At a high sequencing depth, the frequencies of prominent haplotypes are highly stable across varying sample sizes and may be compared directly without subsampling. Note that the variance of a proportion *p* is given by *Var(p) = p ·* (1 − *p*)/*n*, where *n* is the sample size.

### 3.6. No Rare Haplotypes

Let us review a quasispecies composed of a master haplotype at 90% and 10 other haplotypes at 1% each, without any singletons or rare haplotypes. Our aim is to estimate haplotype frequencies by repeated subsampling. A quasispecies without haplotypes at very low frequencies will give approximately the same results with subsampling or not, and both subsampling methods will retrieve very similar results (Table 9 and Table 10).

### 3.7. Flat Quasispecies

Let us consider the case of a quasispecies with n haplotypes, where all have an equal number of reads, *k*, growing from 1 to 10 each. As the frequency of each haplotype increases, they become less rare. Given the number of reads per haplotype, *k*, the probability to sample a given haplotype with replacement after a full resampling cycle of *n · k* random extractions, that is, a bootstrap cycle, is given by 1 − (1 − *k/*(*n · k*))*^(n·k)^ =* 1 − (1 − 1*/n*)*^(n·k)^*, where *n · k* is the sample size. The probability, which, in the limit as n goes to infinity, is 1 − (1/*e*)*^k^*. Table 11 and Figure 4 show the values computed for n = 1000 haplotypes, *k* from 1 to 10 reads each, the computed probability, and the corresponding limits.

In subsampling a fraction *f* of the full sample size, as described in Table 12, the expected number of haplotypes subsampling without replacement is given by the rarefaction equation, which, when applied to this case, results in Equation (1).
(1)$$E_{1}\left[n|k,f\right]=n-n\cdot \left(\begin{array}{c} n\cdot k-k\\ n\cdot k\cdot f \end{array}\right)/\left(\begin{array}{c} n\cdot k\\ n\cdot k\cdot f \end{array}\right)$$

The expected number of haplotypes when subsampling with replacement is given by Equation (2)
(2)$$E_{2}[n|k,f]=1-{\left(1-1/n\right)}^{n\cdot k\cdot f}$$

The results of Equation (1) for *n* = 10,000 haplotypes with frequencies, *k* = 1, 2, …, 10 reads, and subsample fractions, *f* = 0.1, 0.2, …, 1 are represented in Figure 5.

The ratio *E_2_[n|k,f]/E_1_[n|k,f]* gives the fraction of haplotypes estimated in subsampling with replacement with respect to those estimated in subsampling without replacement (rarefaction). This ratio gives a representation of the accuracy obtained in subsampling with replacement in this scenario, and is represented in Figure 6, computed for *n* = 10,000 haplotypes, *k* = 1, 2, …, 10 reads per haplotype, and *f* = 0.1, 0.2, …, 1.

This represents an extreme case with all haplotypes at equivalent and very low frequencies, with no prominent or dominant haplotypes. This is a useful example to show the implications of low-level abundance filters in our context. This type of low-level abundance filter may aim at limiting technical and instrumental noise, but it is not always advisable.

When comparing samples with the rarest haplotypes being excluded by setting a minimum abundance threshold, i.e., such as a minimum of 5 reads per haplotype, the outcomes of subsampling under both methods will tend to be similar, particularly in not-so-extreme cases, such as flat-like quasispecies.

## 4. Discussion

The determination of quasispecies diversity is significantly influenced by sample size, where larger samples generally yield a more accurate estimation. It is well-established that diversity estimation is dependent on the sampling effort, affecting some indices more than others. The higher the effort, the bigger the chances to sample lower frequent and rare haplotypes, an inherent characteristic of quasispecies.
(3)$${\;\;}^{q}D\left(p\right)={\left(\sum_{i=1}^{H}p_{i}^{q}\right)}^{1/\left(1-q\right)}$$

When computing diversity through Hill numbers, *^q^D(p)* (Equation (3)) [25,26], of different orders *q*, they will be limited above by *^0^D*, being the number of haplotypes, and below by *^∞^D*, being the inverse of the frequency of the dominant haplotype. As the order *q* increases, the relative weight of low frequency and rare haplotypes in the computation decreases, as low-frequency values are more heavily affected by the exponent. At *q* = 0, all haplotypes have equal weight regardless of their frequency, while at *q* = ∞, only the highest frequency holds significance. This observation suggests that the sensitivity or dependence of a Hill number with respect to sample size decreases as *q* gets bigger. Considering the correspondence between Hill numbers and other classical diversity indices, we may set the sensitivity order as:*Number of haplotypes* > *Shannon entropy* > *Gini index* > *Master frequency*

The rare haplotype load (RHL) [27] and the quasispecies fitness fractions (QFF) [23] are specific quasispecies diversity indices calculated as fractions of aggregated reads corresponding to haplotypes with frequencies between defined limits. These indices represent prominent fractions, which are relatively insensitive to sample size. Even the fraction of rare haplotypes, computed below 1% or below 0.1%, proves to be robust to sample size, provided that the coverage is high enough to capture with sufficient depth of these fractions.

The number of haplotypes and the related incidence indices [26], such as the number of polymorphic sites or the number of mutations, on one hand, and the Shannon entropy on the other will benefit from rarefaction, and repeated subsampling without replacement to the reference size is required. Higher order diversity indices like the Gini–Simpson index will be less sensitive to sample size and may be rarefied with replacement, when required. The frequency of prominent haplotypes remains highly robust to sample size variations and can be directly compared or rarefied with replacement if needed.

As mentioned earlier, it is advisable to employ subsampling without replacement. In specific cases, such as when examining quasispecies diversity in hepatitis E virus (HEV) [23], this approach is particularly crucial. HEV, characterized by a high mutation rate, exhibits considerable diversity, resulting in haplotypes at very low frequencies. Subsampling with replacement might not fairly represent these low-frequency haplotypes. This especially needs to be considered when analyzing HEV samples from chronic patients treated with ribavirin due to the increase in mutation rates introduced, which leads to a more unstructured quasispecies, leading to even more low-frequency haplotypes [23,28]. On the contrary, for viruses with a less diverse quasispecies, like SARS-CoV-2 [23], as demonstrated earlier in the ‘Flat Quasispecies’ case, subsampling could be effectively carried out through subsampling with replacement. This is supported by the removal of lowest-frequency haplotypes, which allows for the retrieval of comparable results to subsampling without replacement.

In NGS data, haplotype frequencies are obtained as read counts, which are integers. However, when computing diversity indices, frequencies are used as fractions of haplotype reads relative to the sample size (total number of reads), *N*. This introduces granularity in the results, as not all frequencies from 0 to 1 are possible, with the granularity determined by the frequency of a single read, 1/*N*. With a coverage of 1000 reads, the resolution is 0.1%. This resolution may be insufficient for quasispecies analysis, especially when rare haplotypes are of interest, in which a target depth above 100,000 reads per sample/amplicon is recommended.

Controversy in the metagenomics field, mentioned in the introduction, also arises because there are different approaches other than rarefaction when differential abundance analysis is the main objective instead of diversity comparisons. Methods based in counts instead of frequencies, like generalized linear models (GLM), with family distributions like the negative binomial [29], are being used in RNAseq or in label-free proteomics by LC-MS/MS, among others. These methods are used in differential expression studies, aiming to compare the relative abundances of mRNA or proteins between two biological conditions, and use an implicit normalization by offsets, which allow for complex normalizations [30]. In metagenomics, these and other methods based on compositional data analysis (CoDA) [31,32] are also used in the normalization of microbiome abundance tables [10,33]. Nevertheless, when comparing diversity metrics between unbalanced samples, rarefaction is a necessity [34], especially with diversity indices, such as the number of haplotypes, polymorphic sites, mutation frequency, Shannon entropy, Hill numbers, and others, or metrics, which show a dependency of the sample size.

This study lays the groundwork for determining the most appropriate subsampling approach depending on quasispecies structure and the specific indices to be compared. In summary, with high depths, frequencies of prominent haplotypes and fractions of reads are robust to sample size variations and can be compared directly or previously subsampled with replacement. The estimation of Shannon entropy, where low-frequency and rare haplotypes still have a significant role, requires rarefaction by subsampling without replacement to balance biases. The estimation of the number of haplotypes, incidence indices, the fraction of singletons, or of the lowest frequency haplotypes requires subsampling without replacement. As part of the experimental design, a minimum coverage must be established beforehand to reject or repeat any samples falling below that threshold. This minimum coverage sets the level of information conveyed by the study.

## Figures and Tables

**Figure 1 viruses-16-00710-f001:**
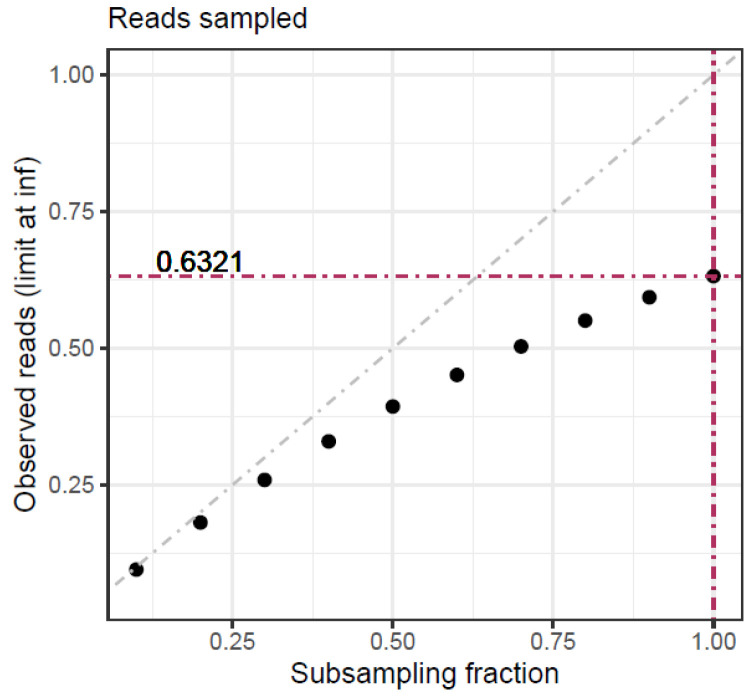
Subsampling with replacement. Theoretical limit to the number of observed items when subsampling with replacement at different subsampling fractions.

**Figure 2 viruses-16-00710-f002:**
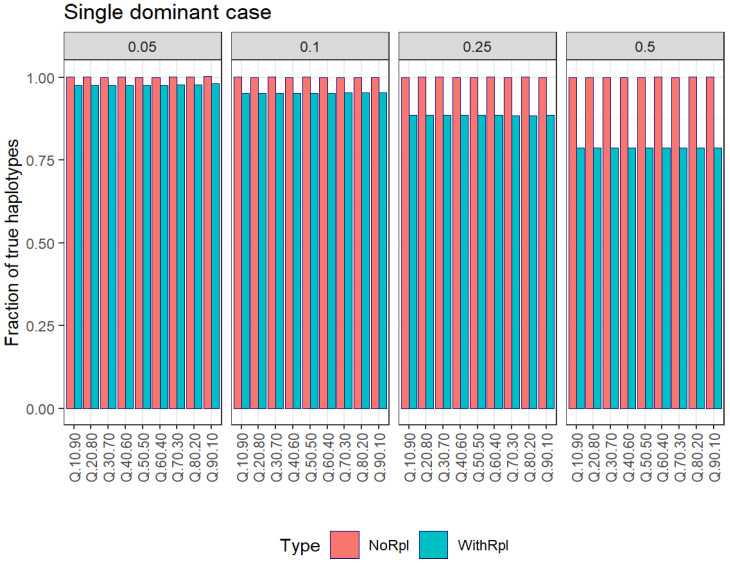
Single-dominant case. Estimation of the number of haplotypes at different subsampling fractions after B resampling cycles with and without replacement.

**Figure 3 viruses-16-00710-f003:**
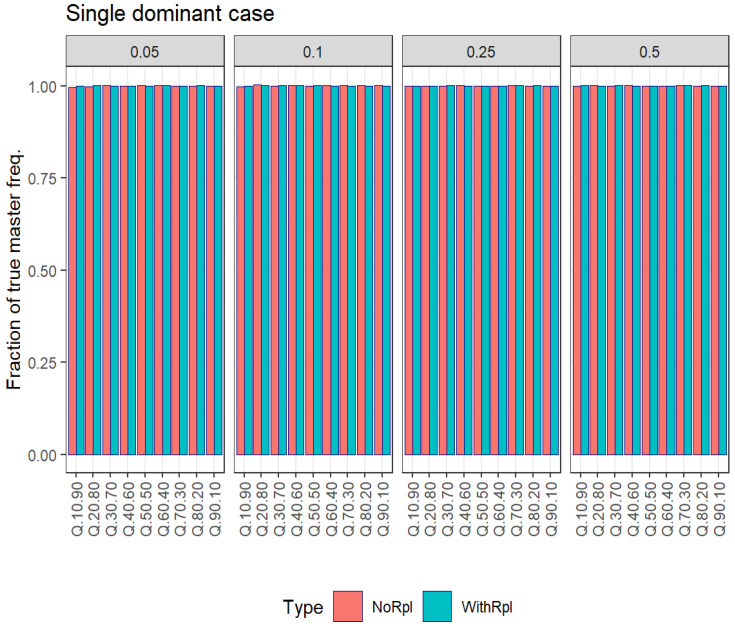
Single-dominant case. Estimation of the master frequencies at different subsampling fractions after B resampling cycles with and without replacement.

**Figure 4 viruses-16-00710-f004:**
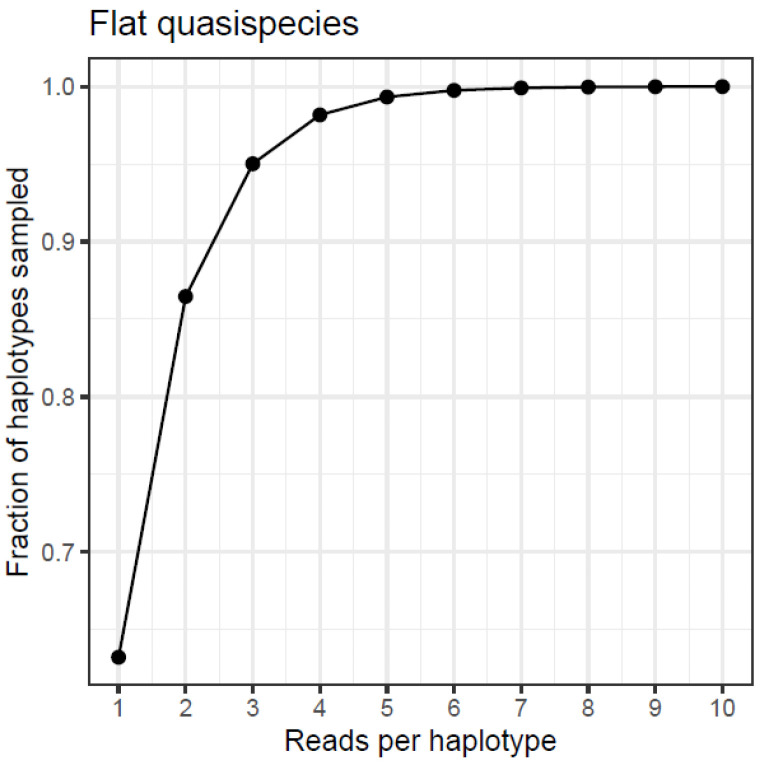
Theoretical limit to the number of observed haplotypes in a bootstrap resample cycle. Flat quasispecies with growing reads per haplotype.

**Figure 5 viruses-16-00710-f005:**
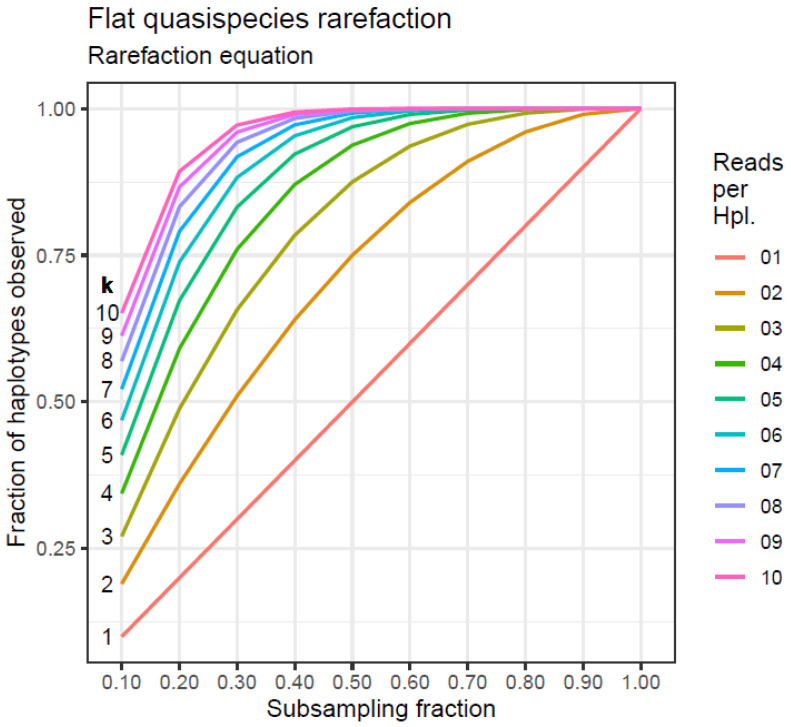
Flat quasispecies rarefaction.

**Figure 6 viruses-16-00710-f006:**
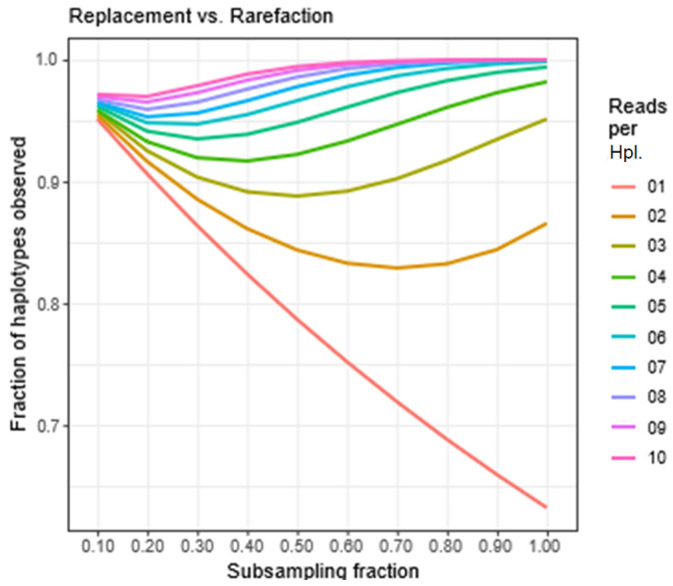
Flat quasispecies. Ratio of number of haplotypes estimated in subsampling with replacement versus those estimated by the rarefaction equation.

**Table 1 viruses-16-00710-t001:** Subsampling a given fraction with replacement. Proportion of items seen and unseen in a single resampling cycle.

Fraction	Seen	Unseen
0.1	0.0952	0.9048
0.2	0.1813	0.8187
0.3	0.2592	0.7408
0.4	0.3297	0.6703
0.5	0.3935	0.6065
0.6	0.4512	0.5488
0.7	0.5034	0.4966
0.8	0.5507	0.4493
0.9	0.5934	0.4066
1.0	0.6321	0.3679

**Table 2 viruses-16-00710-t002:** All-singleton case. Estimating the number of haplotypes. Subsampling a given fraction with replacement.

Frac	True	Expected	Median	IQR	SD	Unique	Replicated
0.1	1000	952.1	952.0	8.00	6.21	0.9520	0.0480
0.2	2000	1813.5	1812.0	17.00	12.52	0.9060	0.0940
0.3	3000	2592.9	2593.0	23.00	15.88	0.8643	0.1357
0.4	4000	3298.1	3295.0	25.50	20.20	0.8238	0.1762
0.5	5000	3936.2	3932.0	33.00	24.95	0.7864	0.2136
0.6	6000	4513.5	4512.0	39.25	27.85	0.7520	0.2480
0.7	7000	5035.9	5033.0	37.00	28.11	0.7190	0.2810
0.8	8000	5508.5	5505.0	42.00	30.37	0.6881	0.3119
0.9	9000	5936.1	5934.0	43.25	32.02	0.6593	0.3407
1.0	10,000	6323.0	6321.5	42.00	32.47	0.6322	0.3678

**Table 3 viruses-16-00710-t003:** Single-dominant quasispecies data.

ID	Master	Hpl. No.
Q.90.10	0.9	10,001
Q.80.20	0.8	20,001
Q.70.30	0.7	30,001
Q.60.40	0.6	40,001
Q.50.50	0.5	50,001
Q.40.60	0.4	60,001
Q.30.70	0.3	70,001
Q.20.80	0.2	80,001
Q.10.90	0.1	90,001

**Table 4 viruses-16-00710-t004:** Single dominant. Estimating number of haplotypes. Median values.

ID	Subsz	NoRpl	WithRpl	Exact
Q.90.10	0.50	5002.0	3933.0	5000
Q.90.10	0.25	2500.0	2215.0	2500
Q.90.10	0.10	1000.0	953.0	1000
Q.90.10	0.05	501.0	490.0	500
Q.80.20	0.50	10,002.5	7866.0	10,000
Q.80.20	0.25	5002.5	4423.0	5000
Q.80.20	0.10	1999.0	1906.0	2000
Q.80.20	0.05	1001.5	977.0	1000
Q.70.30	0.50	14,996.0	11,799.5	15,000
Q.70.30	0.25	7495.5	6635.0	7500
Q.70.30	0.10	2999.0	2858.0	3000
Q.70.30	0.05	1501.0	1466.5	1500
Q.60.40	0.50	20,005.0	15,741.0	20,000
Q.60.40	0.25	10,001.0	8852.0	10,000
Q.60.40	0.10	3999.5	3807.5	4000
Q.60.40	0.05	1998.0	1951.0	2000
Q.50.50	0.50	25,001.0	19,676.5	25,000
Q.50.50	0.25	12,500.5	11,070.0	12,500
Q.50.50	0.10	5006.0	4759.0	5000
Q.50.50	0.05	2499.0	2440.0	2500
Q.40.60	0.50	29,996.0	23,609.5	30,000
Q.40.60	0.25	14,993.0	13,274.0	15,000
Q.40.60	0.10	6000.0	5706.0	6000
Q.40.60	0.05	3001.5	2927.5	3000
Q.30.70	0.50	35,001.0	27,542.5	35,000
Q.30.70	0.25	17,504.5	15,487.5	17,500
Q.30.70	0.10	7004.0	6661.0	7000
Q.30.70	0.05	3499.0	3415.0	3500
Q.20.80	0.50	39,997.0	31,477.5	40,000
Q.20.80	0.25	20,002.5	17,701.0	20,000
Q.20.80	0.10	7997.0	7613.5	8000
Q.20.80	0.05	4003.0	3904.0	4000
Q.10.90	0.50	45,001.0	35,409.0	45,000
Q.10.90	0.25	22,502.0	19,914.0	22,500
Q.10.90	0.10	9003.0	8565.0	9000
Q.10.90	0.05	4503.0	4389.0	4500

**Table 5 viruses-16-00710-t005:** Single dominant. Estimating master frequency. Median values.

ID	Subsz	NoRpl	WithRpl	Exact
Q.90.10	0.50	0.899980	0.90005	0.9
Q.90.10	0.25	0.900040	0.90004	0.9
Q.90.10	0.10	0.900100	0.90000	0.9
Q.90.10	0.05	0.900000	0.90000	0.9
Q.80.20	0.50	0.799970	0.80010	0.8
Q.80.20	0.25	0.799940	0.80014	0.8
Q.80.20	0.10	0.800200	0.79980	0.8
Q.80.20	0.05	0.799900	0.80020	0.8
Q.70.30	0.50	0.700100	0.70016	0.7
Q.70.30	0.25	0.700220	0.70012	0.7
Q.70.30	0.10	0.700200	0.69980	0.7
Q.70.30	0.05	0.700000	0.69980	0.7
Q.60.40	0.50	0.599920	0.60004	0.6
Q.60.40	0.25	0.600000	0.59988	0.6
Q.60.40	0.10	0.600150	0.59990	0.6
Q.60.40	0.05	0.600600	0.60040	0.6
Q.50.50	0.50	0.500000	0.49985	0.5
Q.50.50	0.25	0.500020	0.49982	0.5
Q.50.50	0.10	0.499500	0.50005	0.5
Q.50.50	0.05	0.500400	0.50000	0.5
Q.40.60	0.50	0.400100	0.39998	0.4
Q.40.60	0.25	0.400320	0.39988	0.4
Q.40.60	0.10	0.400100	0.40040	0.4
Q.40.60	0.05	0.399900	0.39980	0.4
Q.30.70	0.50	0.299993	0.30011	0.3
Q.30.70	0.25	0.299860	0.30008	0.3
Q.30.70	0.10	0.299700	0.30010	0.3
Q.30.70	0.05	0.300400	0.30000	0.3
Q.20.80	0.50	0.200072	0.19978	0.2
Q.20.80	0.25	0.199924	0.19992	0.2
Q.20.80	0.10	0.200400	0.20025	0.2
Q.20.80	0.05	0.199600	0.20020	0.2
Q.10.90	0.50	0.100000	0.10007	0.1
Q.10.90	0.25	0.099960	0.09990	0.1
Q.10.90	0.10	0.099800	0.10000	0.1
Q.10.90	0.05	0.099600	0.10000	0.1

**Table 6 viruses-16-00710-t006:** Prominent haplotypes, quasispecies composition.

Number of Reads	100,000
Number of haplotypes	3083
Prominent haplotypes (read counts)	49,231, 24,615, 12,308, 6154, 3077, 1538
Singletons (reads)	3077

**Table 7 viruses-16-00710-t007:** Prominent haplotypes. Subsampling without replacement. Median values.

Subs	SngFr	Hpl_1	Hpl_2	Hpl_3	Hpl_4	Hpl_5	Hpl_6	Ov1
True	0.03077	0.49231	0.24615	0.12308	0.06154	0.03077	0.01538	0
0.5	0.03076	0.49211	0.24626	0.12315	0.06148	0.03084	0.01542	0
0.25	0.03080	0.49224	0.24606	0.12316	0.06164	0.03076	0.01536	0
0.1	0.03090	0.49240	0.24635	0.12280	0.06160	0.03070	0.01540	0
0.05	0.03100	0.49280	0.24620	0.12280	0.06120	0.03060	0.01520	0

**Table 8 viruses-16-00710-t008:** Prominent haplotypes. Subsampling with replacement. Median values.

Subs	SngFr	Hpl_1	Hpl_2	Hpl_3	Hpl_4	Hpl_5	Hpl_6	Ov1
True	0.03077	0.49231	0.24615	0.12308	0.06154	0.03077	0.01538	0.00000
0.5	0.01872	0.49230	0.24604	0.12302	0.06146	0.03078	0.01542	0.01210
0.25	0.02396	0.49232	0.24626	0.12308	0.06148	0.03068	0.01536	0.00684
0.1	0.02780	0.49215	0.24620	0.12320	0.06170	0.03070	0.01540	0.00285
0.05	0.02900	0.49280	0.24600	0.12320	0.06140	0.03060	0.01540	0.00140

**Table 9 viruses-16-00710-t009:** No rare haplotypes. Subsampling without replacement.

Subs	HplNo	Hpl_01	Hpl_02	Hpl_03	Hpl_04	Hpl_05
True	11	0.90000	0.01000	0.01000	0.01000	0.0100
0.5	11	0.89996	0.01002	0.01004	0.01002	0.0100
0.25	11	0.89990	0.01000	0.01000	0.01004	0.0100
0.1	11	0.90000	0.01010	0.01000	0.01000	0.0101
0.05	11	0.90000	0.01000	0.01000	0.01000	0.0100
Subs	Hpl_06	Hpl_07	Hpl_08	Hpl_09	Hpl_10	Hpl_11
True	0.0100	0.01000	0.01000	0.01	0.01000	0.01000
0.5	0.0100	0.01002	0.01001	0.01	0.01002	0.00999
0.25	0.0100	0.00996	0.00996	0.01	0.01000	0.00996
0.1	0.0101	0.01000	0.01010	0.01	0.01000	0.01000
0.05	0.0100	0.01000	0.01000	0.01	0.01000	0.01000

**Table 10 viruses-16-00710-t010:** No rare haplotypes. Subsampling with replacement.

Subs	HplNo	Hpl_01	Hpl_02	Hpl_03	Hpl_04	Hpl_05
True	11	0.90000	0.01000	0.01000	0.01000	0.01000
0.5	11	0.90006	0.00999	0.01002	0.01002	0.01004
0.25	11	0.90004	0.01000	0.01004	0.00992	0.01004
0.1	11	0.90010	0.01000	0.01000	0.01000	0.01000
0.05	11	0.90010	0.01000	0.01000	0.01000	0.01020
**Subs**	**Hpl_06**	**Hpl_07**	**Hpl_08**	**Hpl_09**	**Hpl_10**	**Hpl_11**
True	0.01000	0.01000	0.01000	0.01000	0.01000	0.01000
0.5	0.01002	0.00998	0.01002	0.00998	0.00996	0.01002
0.25	0.01000	0.01000	0.00994	0.01004	0.01004	0.01000
0.1	0.00990	0.01000	0.01000	0.01000	0.01000	0.01000
0.05	0.00980	0.00980	0.01000	0.00980	0.00990	0.01000

**Table 11 viruses-16-00710-t011:** Flat quasispecies: full bootstrap cycle results at growing haplotype frequencies to this case results in Equation (1).

nHpl	k	Reads	Prob	Limit
1000	1	1000	0.6323046	0.6321206
1000	2	2000	0.8648001	0.8646647
1000	3	3000	0.9502876	0.9502129
1000	4	4000	0.9817210	0.9816844
1000	5	5000	0.9932789	0.9932621
1000	6	6000	0.9975287	0.9975212
1000	7	7000	0.9990913	0.9990881
1000	8	8000	0.9996659	0.9996645
1000	9	9000	0.9998771	0.9998766
1000	10	10,000	0.9999548	0.9999546

**Table 12 viruses-16-00710-t012:** Flat quasispecies subsampling.

Haplotypes	*n*
Reads per haplotype	*k*
Full sample size	*n · k*
Subsampling fraction	*f*
Subsample size	*round*(*n · k · f)*

## Data Availability

Data are contained within the article and Appendix A.

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
