# Peer review of "Virus Quasispecies Rarefaction: Subsampling with or without Replacement?"

_viruses, 2024, doi:10.3390/v16050710_

Round 1

Reviewer 1 Report

Comments and Suggestions for Authors

The manuscript describes artifacts that arise in next generation sequence analysis of diversity of sequences within a virus sample, with emphasis on human viruses with particularly fast variation. Numbers of diversity metrics are analyzed under the conditions that comparing diversity in two samples of different sizes is of interest. Many of these metrics do not scale cleanly with sample size and so a common practice is to reduce the sampling size of the larger sample to make the two samples equal in numbers of sequences. The manuscript describes a bias that occurs if this reduction is done by resampling with replacement, a procedure that is conveniently available because of its involvement in bootstrap analysis, however does not emulate the sampling as it occurs in the NGS methods themselves. As described by example, the bias can be large or small depending on the sample structure and the specific metric involved.

NGS is often used with a mindset of not paying much attention to the details, so it’s a welcome contribution to see someone trying to improve on that. The manuscript proceeds by demonstrating the change in diversity estimates given the two resampling methods using different metrics for several test populations intended to cover the different distributional structures one’s experimental data might be expected to have. In that respect, it is very thorough. In the discussion they also relate the differential performance of the different methods to something they call “sensitivity”. It centers around the obvious concept that you can’t get an accurate measure of diversity of low frequency variants if you are looking at barely enough reads to see them. The discussion of sensitivity is more giving a rule of thumb than a basis for conducting a statistical analysis, but it does indicate that different of the metrics in use may have different power near the sampling threshold on the same data sets. Then, finally, they mention two other methods of analysis that they didn’t demonstrate: GLM and CoDA. I got the impression that these exemplify a more rigorous statistical approach. I didn’t fully understand if the authors were mentioning these for completeness, but weren’t very familiar with them, or if they were intending to recommend these approaches.

I won’t pretend to be as much of an expert on these issues as the authors. I will make suggestions for improving the context for the reader.

1) The abstract could be more clear about what kind of viruses you are talking about.

2) The bulk of the paper implies that the resampling bias problem is an ongoing problem with published papers in this field. But the discussion eventually says that comparison of diversity between samples isn’t commonly done in the literature. There it seems to be more of a potential rising problem. Try to make the context of the problem in the introduction and the discussion match.

3) Resolve my confusion mentioned above about whether you’re recommending GLM and/or CoDA or just mentioning them for completeness.

Author Response

We would like to thank reviewer for their kind and insightful comments on the topic developed in this paper. We are confident that addressing the reviewer’s comments will improve the quality and clarity of our manuscript.

1) The abstract could be more clear about what kind of viruses you are talking about.

Answer: According to reviewer comment, the abstract has been modified indicating that the hypothetical cases refer to different types of quasispecies structures. See line 20-21 of the clean version: “… of quasispecies showing different quasispecies structure in the sense of haplotype genomic composition, …

Also under the Methods section, we have added a description of the type of data used in the computations. Lines 97 to 101 of the clean version: “Intensive resampling simulations under each hypothetical case have been carried out, to support and extend the theoretical considerations for each case and resampling scheme. Each described example is represented as a vector of reads, where each element corresponds to a different haplotype, and the total coverage is given by the sum of vector elements.”

2) The bulk of the paper implies that the resampling bias problem is an ongoing problem with published papers in this field. But the discussion eventually says that comparison of diversity between samples isn’t commonly done in the literature. There it seems to be more of a potential rising problem. Try to make the context of the problem in the introduction and the discussion match.

Answer: Thank you for the comment. We have realized that the paragraph added for completeness in the discussion caused confusion, and it has been fully rewritten. See lines 367 to 380 in the clean version: The controversy in the metagenomics field, mentioned in the introduction, arises also because there are different approaches other than rarefaction when differential abundance analysis is the main objective, instead of diversity comparisons. Methods based in counts instead of frequencies, like generalized linear models (GLM), with family distributions like the Negative Binomial (29), are being used in RNAseq or in label-free proteomics by LC-MS/MS, among others. These methods are used in differential expression studies, aiming to compare relative abundances of mRNA or proteins between two biological cond-tions, and use an implicit normalization by offsets, which allow for complex normalizations (30). In metagenomics, these and other methods based on compositional data analysis (CoDA) (31,32) are also used in the normalization of microbiome abundance tables (10,33). Nevertheless, when comparing diversity metrics between unbalanced samples, rarefaction is a necessity (34). Specially with diversity indices such as the number of haplotypes, polymorphic sites, mutation frequency, Shannon entropy, Hill numbers, and others, or metrics which show a dependency of the sample size..”

3) Resolve my confusion mentioned above about whether you’re recommending GLM and/or CoDA or just mentioning them for completeness.

Answer: Mentions to GLM and CoDA have been added for completeness in discussion (see lines 367 to 380 in the clean version), relative to the controversy in the metagenomics field that has been commented in the introduction (lines 49 to 52 in the clean version).

Reviewer 2 Report

Comments and Suggestions for Authors

The topic of the manuscript is of interest. The study aims to provide a comprehensive examination of resampling schemes through various hypothetical cases to elucidate their implications in general scenarios. The review highlights the challenge caused by the sensitivity of certain diversity indices to sample size variations and underscores the significance of rarefaction as a tool to address this issue. However, some minor points can be improved,

1.      It is not clear the utilized reads are from which source genomes; this should be explicitly described correlation with examples of virus data that could fit with journal’s objectives?

2.      Page 1, line 33, give the full name of HCV.

3.      Lines 41 and 42 lack of citation.

4.      Line 95, it says “All analyses”, which analyses, describe briefly?

5.      Method section is very short. Box 1 should contain citation of each detail.

6.      Line 229, it should be in a table.

Author Response

  1. It is not clear the utilized reads are from which source genomes; this should be explicitly described correlation with examples of virus data that could fit with journal’s objectives?

Answer: Thank you for your insightful comment. We have inserted a paragraph in Methods section describing in more detail the methodology used in this study. See lines 97-106 in the clean version: Intensive resampling simulations under each hypothetical case have been carried out, to support and extend the theoretical considerations for each case and resampling scheme. Each described example is represented as a vector of reads, where each element corresponds to a different haplotype, and the total coverage is given by the sum of vector elements. Based on this representation of quasispecies composition for each case, all computations were conducted using R (16), with the help of packages knitr (17), tidyverse (18), ggplot2 (19), and dqrng (20). Resampling without replacement was optimized using the dqsample.int function from the package dqrng (20), and with the help of a full sample size vector mapping reads to haplotypes. The full R code used is given in Appendix A (Supplementary material).”

  1. Page 1, line 33, give the full name of HCV.

Answer: Done. See lines 34 to 35 in the clean version

  1. Lines 41 and 42 lack of citation.

Answer: Thank you for the observation. We have added three citations in line 43 and line 45 of the clean version: (6) Martell et al 1992, as the first study describing hepatitis C viral as quasispecies using clonal approach; (7) Gregori et al 2014, in which we compared NGS with clonal studies; and in the next sentence we have added citation (8) Gregori et al 2013, in which we described the amplicon-based studies using deep-sequencing.

  1. Line 95, it says “All analyses”, which analyses, describe briefly?

Answer: Thank you. We have clarified the type of analysis performed (see lines 97-106 in the clean version).

  1. Method section is very short. Box 1 should contain citation of each detail.

Answer: Thank you. We have extended the explanation of the methodological procedure (see lines 97 to 106 in the clean version). Citations have been added in legend of Box 1 (see line 107 in the clean version).

  1. Line 229, it should be in a table.

Answer: Thank you for the comment. We have made Table 6 (see between lines 238 to 239 of the clean version) and renamed the rest of the tables in order of appearance in the manuscript.

Reviewer 3 Report

Comments and Suggestions for Authors

The study/review is unique in that it sets ground to determine the most appropriate subsampling approach and highlights the importance of subsampling without replacement. The study adds to this area of microbiome/virome research and explains in detail the effects of subsampling with and without replacement. A couple of minor comments:

Line 33 define HCV

Line 39 define NGS and remove definition from line 343

Box 1: The box can benefit from having headers. Is there a purpose of having some sentences starting with capital letters?

Line 243 can the authors expand on any study looking at different sequencing depth and noted that this is the case for deeply sequenced samples?

Comments on the Quality of English Language

English is good 

Author Response

Line 33 define HCV

Answer: Done, thank you (see lines 34-35 in the clean version).

Line 39 define NGS and remove definition from line 343

Answer: Done, thank you (see line 44 in the clean version).

Box 1: The box can benefit from having headers. Is there a purpose of having some sentences starting with capital letters?

Answer: Thank you. We have added a header and homogenized the text with capital letters at the beginning of each sentence (see Box1 at page 3, between lines 107 to 108 in the clean version).

Line 243 can the authors expand on any study looking at different sequencing depth and noted that this is the case for deeply sequenced samples?

Answer:  Thank you for the comment. We have added in lines 251-252 in the clean version, a sentence in which we have quoted a paper that describe differences in isolates with different sequencing depths. We observed similar results with real Hepatitis E Virus data (23) sequencing sample replicates at different depths (24).” Moreover, we have added further explanation in lines 254-256 in the clean version. “Note that the variance of a proportion p is given by Var(p) = p · (1 − p)/n, where n is the sample size.”